# Understanding Impacts of SNAP Fruit and Vegetable Incentive Program at Farmers’ Markets: Findings from a 13 State RCT

**DOI:** 10.3390/ijerph19127443

**Published:** 2022-06-17

**Authors:** Allison Karpyn, Julia Pon, Sara B. Grajeda, Rui Wang, Kathryn E. Merritt, Tara Tracy, Henry May, Ginnie Sawyer-Morris, McKenna M. Halverson, Alan Hunt

**Affiliations:** 1Center for Research in Education and Social Policy, University of Delaware, Newark, DE 19713, USA; tetracy@udel.edu (T.T.); hmay@udel.edu (H.M.); gsmorris@udel.edu (G.S.-M.); mhalvers@udel.edu (M.M.H.); 2Target, Mountain View, CA 94040, USA; julia.a.pon@gmail.com; 3Podium, Orem, UT 84097, USA; saragrajeda28@gmail.com; 4Mathematica, Washington, DC 20002, USA; ruiwang@udel.edu; 5Robert Half, San Francisco, CA 94111, USA; merrittkt@gmail.com; 6Musconetcong Watershed Association, Asbury, NJ 08802, USA; alan@musconetcong.org

**Keywords:** farmers’ markets, fruits and vegetables, incentive programs, Supplemental Nutrition Assistance Program (SNAP), low-income, food security

## Abstract

Disparities in healthy food access and consumption are a major public health concern. This study reports the findings from a two-year randomized control trial conducted at 77 farmers’ markets (FMs) in 13 states and the District of Columbia that sought to understand the impact of fruit and vegetable (FV) incentive vouchers, randomly issued at varied incentive levels to Supplemental Nutrition Assistance Program (SNAP) recipients, for use at FMs. Measures included FV and overall household food purchasing; FV consumption; food insecurity; health status; market expenditure; and demographics. A repeated-measures mixed-effects analysis and the Complier Average Causal Effect (CACE) were used to examine outcomes. Despite 82% reporting food insecurity in the prior year, the findings showed that financial incentives at FMs had statistically significant, positive effects on FV consumption; market expenditures increased with added incentives. SNAP recipients receiving an incentive of USD 0.40 for every USD 1.00 in SNAP spent an average of USD 19.03 per transaction, while those receiving USD 2 for every USD 1 (2:1) spent an average of USD 36.28 per transaction. The data showed that the incentive program at the highest level (2:1) maximally increased SNAP FM expenditure and FV consumption, increasing the latter by 0.31 daily cups among those who used their incentive (CACE model).

## 1. Introduction

Disparities in healthy food access are prevalent in the United States and contribute to the disproportionately low intake of fruits and vegetables (FVs) among low-income populations [1]. The Supplemental Nutrition Assistance Program (SNAP) aims to mitigate this disparity by providing participants with gross incomes at or below 130% of the federal income poverty line with monetary benefits for food [2]. Although research demonstrates that SNAP is effective in reducing rates of food insecurity, many participants still struggle to afford FVs. The inaccessibility of FVs is problematic, as low FV intake is associated with an increased risk for a host of chronic diseases, including diabetes, cardiovascular disease, obesity, and cancer [3]. Therefore, research is needed to evaluate strategies that increase the purchasing and consumption of FVs among low-income populations.

Farmers’ markets (FMs) play a vital role in advancing equitable food systems and fostering community resilience [4]. The implementation of FV incentive programs at community-based FMs is a growing strategy being leveraged to increase SNAP shoppers’ access to healthy, nutritious food. By supplementing the monetary value of SNAP, these incentives reduce barriers to FV access that are common among low-income populations, such as cost and availability [5,6]. A growing body of research demonstrates the utility of incentive programs for increasing SNAP participants’ expenditures on and intake of FVs at FMs [7,8,9,10].

Although numerous studies have examined the effectiveness of incentive programs for increasing SNAP participants’ purchasing power to spend on FVs at FMs, many of these studies have relied on small, geographically limited samples as well as self-reported measures of FV expenditures. Therefore, the goal of this study was to examine the causal impact of selected (and incrementally different) incentive program innovations, both financial and nonfinancial, on SNAP customers’ purchases of FVs at FMs across the U.S. The study goals also included examining the causal impact of these incentives on participants’ overall and FV grocery purchasing and consumption of FVs.

## 2. Materials and Methods

This field-based, coordinated, multi-site randomized controlled trial (RCT) was conducted at 77 FMs in 13 states and the District of Columbia (DC). A total of 91 FMs signed up for the RCT; however, 14 FMs did not recruit participants. Wholesome Wave, an organization that supports FMs via staffing, financial, and technical assistance, operated the FMs observed in this study. SNAP-eligible customers were the focus of the present study, although all FMs served both non-SNAP and SNAP-eligible customers.

### 2.1. Participants and Recruitment

Between September 2015 and October 2017, FM managers and volunteers recruited SNAP-eligible customers on-site via flyers in both Spanish and English. Participants were eligible for the study if they used SNAP Electronic Benefit Transfer at the FMs and were at least 18 years of age. SNAP shoppers who expressed interest in the study were asked to fill out a numbered enrollment “ticket”, which inquired about their contact information (i.e., name, email address, and cell phone number). FM managers collected these enrollment tickets during each FM operating season, typically lasting about four to six months, and mailed the tickets to the research partner for formal study enrollment.

After each data collection period, FM managers sent completed enrollment tickets to the research team, who then sent participants an electronic link via text or email containing the Qualtrics^TM^ survey. Data were collected over five periods, each of which lasted approximately four to six months. Participants could sign up on a rolling basis and completed surveys at the start of the month they received their incentive and again at the start of the subsequent month, when they would be randomly assigned a different incentive amount. Incentive levels were not the same for every market and were intentionally varied as part of the research design and due to implementation considerations. For example, some markets had pre-existing dollar-for-dollar matching programs and did not want to reduce benefits to less than one dollar for some customers. A detailed description of incentive levels and market types is provided in a subsequent section of this paper. To maintain consistency across the study, however, all markets had only 3 possible incentive levels, as described later in the incentive levels section. Participants who completed the survey were provided with an FM incentive that aligned with the FM standards for their market, ranging between USD 0.40 and USD 2 per USD 1 SNAP spent. Participants who declined the survey invitation remained eligible to receive other incentive programs operating at their FM.

Shoppers who responded to the invitation and who both consented to participate and completed the corresponding online survey were randomly awarded one of three monetary incentive amounts or a nonmonetary incentive. Incentives were authorized for use at the participant’s primary FM or at another allowable FM within their network for the remainder of the month, at which point the monetary incentive expired. At the start of the next month, participants were invited to complete another survey if they chose to remain in the study. Continued participation required completion of the survey again, after which an additional randomized incentive was assigned. Finally, at the end of each round (roughly an FM season), we included a final ‘follow-up’ month, during which prior participants were invited to complete the survey with the same odds of winning randomly assigned incentives as in prior months. No new RCT participants were recruited during this follow-up month.

This study and its procedures were approved by the University of Delaware Institutional Review Board. Informed consent was obtained from all participants, and the survey included a statement regarding the details of the research and voluntary participation. Study materials were made available in English and Spanish.

### 2.2. Incentive Levels

After enrollment and baseline survey completion, participants were randomly assigned (computer-generated) to one of three conditions: (1) no additional monetary incentive beyond baseline; (2) moderate monetary incentive; or (3) the highest monetary incentive. During certain times in the RCT and at certain FMs, a nonmonetary incentive (i.e., a reusable grocery bag imprinted with a healthy eating message) was randomly assigned as a fourth option. When awarded, the nonmonetary incentive was given only once during the month; however, the recipient remained eligible to receive their FM’s baseline monetary incentive throughout the entire month.

Levels of incentives were determined on the basis of the FM where the participant shopped. That said, the participant was randomly assigned to an incentive level that could vary from month to month. Each FM always had the same three possible levels of incentives, which depended on the FM where the participant reported shopping, and the participant was eligible for one of that FM’s three incentive amounts. FM incentive amounts included 2 groupings of markets (type A and B). FM Type A was characterized by: USD 1 (spent) received USD 0.40 additional (1:0.4 baseline); USD 1 (spent) received USD 0.80 (1:0.8 moderate); or, USD 1 (spent) received USD 1.00 (1:1.0 highest); FM Type B: USD 1 (spent) received USD 1 additional (1:1 baseline); USD 1 (spent) received USD 1.50 (1:1.5 moderate); or, USD 1 (spent) received USD 2.00 (1:2 highest). In all cases, participants could utilize their monetary incentives only on FVs.

### 2.3. Measures

A Qualtrics^TM^ survey was used to assess participant demographic information, FV purchasing and consumption, and food insecurity. US Census parameters were used as a basis for measuring participant demographic characteristics, including race, ethnicity, gender, age, family size, and region. SNAP eligibility was used as a proxy for low-income status.

To assess participants’ FV consumption, 10 dietary recall items from the Dietary Screener Questionnaire (DSQ) [11] were administered. As a monthly dietary recall, these excerpted questions from the DSQ asked about the frequency of consumption in the past month of selected foods and drinks. The 10 dietary recall items taken from the DSQ also considered fresh FVs as well as FVs bought in prepared forms or from mixed foods (e.g., 100% fruit juices, refried beans, salsa, tomato sauces, french fries, and pizza). Responses to these survey questions were converted to estimates of dietary FV intake provided in cup equivalents and based on a set of scoring algorithms developed by NHANES (2009–2010). The DSQ’s intraclass correlations for test–retest reliability ranged from 0.62 to 0.67 for FVs for men and women combined. These reliabilities are considered adequate and approach the accepted levels (0.7) for research.

One item from the Centers for Disease Control and Prevention’s (CDC) Behavioral Risk Factor Surveillance System Questionnaire (BRFSS) was used to measure participants’ health status [12]. The two-item Hunger Vital Sign food insecurity screener [13], which has been adopted by the U.S. Department of Agriculture for food security assessments, was used to identify families at risk for food insecurity: (1) “Within the past 12 months we worried whether our food would run out before we got money to buy more,” and (2) “Within the past 12 months the food we bought just didn’t last and we didn’t have money to get more.” The two-item food insecurity screener has high sensitivity (97%), specificity (83%), and convergent validity compared with the longer 18-item US Household Food Security Scale used by the Current Population Survey, making it an effective substitute tool to annually monitor food-security status [13].

Participant FV purchases by different incentive levels were collected at the FMs using the FMTracks^TM^ data capture system described elsewhere [14]. These datasets were connected to the survey data to compare the variation in purchase amount by different incentive levels using the participant’s identifier (i.e., the initials of their first and last names plus the last four digits of their unique SNAP card).

### 2.4. Statistical Analysis

A total of 23,291 survey invitations were sent via email or text to both first-time participants and to those who agreed to complete the survey in subsequent months. Of the surveys sent, 30.5% (*n* = 7097) were completed. The number of first-time completers of the survey between September 2015 and October 2017 was 3073. Between September 2015 and October 2017, 5186 enrollment tickets were received from the national sample of FMs participating in the RCT.

Cases were deleted pairwise, such that participants who skipped one question were excluded from the analysis for that question but were still included in the analyses of the other questions for which they provided complete information. Data were coded as missing and excluded from the analysis if participants responded with “unsure” or “prefer not to answer.” As a result, the number of responses for each question ranged from 2661 to 3013.

We conducted a descriptive analysis of the characteristics of SNAP FM customers: SNAP FM participant grocery spending, FV consumption, and health status. We also examined differences in FV purchasing, grocery purchasing, consumption, and related indicators on the basis of the level of the incentive amount received. A one-way ANOVA was used to detect any significant differences in SNAP dollars spent by participants at different incentive levels.

Furthermore, we conducted a repeated-measures mixed-effects analysis to estimate potential changes in outcome variables after participants were assigned an incentive. Regarding FV expenditures, the repeated-measures model used a log transformation of the dollars spent on FVs over the course of the month to account for skewness in the data. In addition, the model controlled for household size since the number of dollars spent is related to the number of people to feed in the household. The Complier Average Causal Effect (CACE) [15] was also calculated for FV consumption, in which a significant finding was identified to adjust the repeated-measures model results to calculate the effects for only those participants who used their randomly assigned incentive. The CACE estimate was calculated by dividing the ITT estimate by the proportion of compliers in the treatment group or those who were assigned to an incentive level and received the incentive they were assigned. All outcomes were examined on the basis of data from SNAP participants who completed a survey once at the beginning of the month and again at the start of the following month.

## 3. Results

Of the 2968 first-time survey respondents who answered the gender questions, 82% (*n* = 2446) were female. The majority of respondents, 64% (*n* = 1851), were between the ages of 18 and 47. Regarding race and ethnicity, 72% (*n* = 1959) of the respondents were White and 18% (*n* = 515) were Hispanic. Table 1 summarizes demographic data and the prevalence of health conditions faced by SNAP shoppers.

At baseline, (*n* = 2956), 82% of FM SNAP shoppers had experienced food insecurity in the prior year. More than one in four (26%) stated that they were in fair or poor health. When asked about health conditions, 13% reported having diabetes, and 23% reported having high blood pressure.

The survey also asked SNAP FM shoppers about the amount spent on all groceries and the amount spent on FVs as part of their overall grocery budget, not just items purchased at the FM. Per month, each household spent, on average, USD 153.76 on FVs. FV purchasing comprised 45% of the total amount spent on groceries. Refer to Table 2 for expenditure data.

The data on FV consumption revealed that SNAP shoppers consumed, on average, 3.00 daily cups of FVs (an amount that included french fries) at baseline. Males consumed 3.27 cups of FVs per day, while females consumed 2.95 cups per day. Overall, the average amount of FVs that adults aged 18–47 consumed was about 3.03 cups per day. Adults aged 48–67 consumed 2.95 cups of FVs per day, and older adults (age 68+) consumed about 2.96 cups of FVs per day. Table 2 summarizes these data.

In total, the study issued 6979 monetary incentives that could be used multiple times over one month at the national sample of FMs. Of these, 3144 incentives were redeemed at least once, and in total, incentives were redeemed during 5253 visits, an average of 1.67 times across the month. Those with an incentive spent an average of USD 34.39 in SNAP funds alone (before any additional incentive was applied) per visit to the FM, as shown in Table 3. Table 4 presents a breakdown of the incentives issued for use at FMs by incentive level, including whether the incentive level was baseline, moderate, or highest for that FM. Table 5 establishes incentive assignments as percentages of the total numbers of incentives awarded throughout the study.

Repeated-measures mixed-effects analysis was conducted to estimate potential changes in outcome variables after participants were assigned an incentive. At the highest incentive only (i.e., USD 2.00 incentive at USD 1.00 baseline Type B FMs), participants consumed a statistically significant higher quantity of FVs (0.16 daily cups) compared with participants receiving the moderate and baseline incentives at the Type B FMs. No other statistically significant differences were found for the other incentive levels, nor for the study’s other outcome variables (i.e., grocery FV expenditures). Table 6 summarizes these outcomes data on the basis of the repeated-measures mixed-effects analysis and reports the data according to incentive level.

The repeated-measures model estimates include all participants, whether or not they used the incentive they were randomly assigned. As noted, the CACE methodology adjusted the repeated-measures model results to calculate the effects for only those participants who used their randomly assigned incentive [15]. According to the CACE calculation, the FV consumption of participants who used their 2.0 incentive increased by 0.31 daily cups, almost twice the average for all participants who were randomly assigned the 2.0 incentive level. With one exception (household size), no significant differences were found for monthly FV expenditures between incentive levels.

Further, no significant differences in FV consumption or purchasing were found for those who received a nonmonetary incentive (a grocery bag).

The data showed a steady increase in the amount of SNAP spent at each incremental incentive level. SNAP recipients who shopped at FMs and who were awarded a baseline incentive of USD 0.40 for every USD 1.00 in SNAP spent an average of USD 19.03 per transaction, while those receiving the highest incentive level of USD 2.00 spent an average of USD 36.28 per transaction. At Type A FMs, participants at both the moderate (0.8) and higher (1.0) incentive levels spent significantly more of their SNAP dollars compared with participants at the lower 0.40 incentive at Type A FMs. At the Type B FMs, where the baseline level incentive was 1.0, the 2.0 level incentive showed a statistically significant increase in SNAP expenditures, as shown in Table 3. In aggregate, across both Type A and Type B FMs, each stepwise increase in incentive level resulted in a statistically significant increase in SNAP expenditures, except between the intermediary USD 1.00 to USD 1.50 levels.

## 4. Discussion

This study provides rigorous evidence about the causal effect of FV incentive use at FMs across a wide range of incentive levels on SNAP participants’ FV consumption and purchasing. Specifically, our findings suggest that financial FV incentives randomly awarded to SNAP shoppers at FMs have statistically significant, positive effects on FV consumption, increasing consumption for those at the highest (2.0) incentive level by 0.16 cups/day. Even stronger positive effects were found when FV consumption was calculated for only those SNAP shoppers who used their incentive: at the 2.0 level, consumption increase almost doubled to 0.31 cups/day.

Our findings are congruent with previous research that has suggested that by increasing SNAP shoppers’ purchasing power, FM incentives increase FV consumption [7,9,10,16,17]. Additionally, our findings showed statistically significant, higher SNAP spending on FVs that incrementally increased with incentive levels. Further, we found no statistically significant effect of the nonmonetary incentive (e.g., grocery bag) in increasing FV consumption or purchasing.

The findings from this study underscore the potential for financial incentives provided to SNAP shoppers at FMs to increase FV consumption. On the basis of these results, we make several recommendations. First, we recommend that a larger-scale test of the highest incentive level (i.e., USD 2 for every USD 1 spent) compared with a very low incentive level (i.e., USD 0.40 or below for every USD 1 spent) be conducted. Our present results, limitations notwithstanding, suggest that where resources allow, higher-level incentives, such as those at USD 2, provide greater benefits to fruit and vegetable intake than those at lower incentive levels.

At baseline, participants reported consuming 2.77 cups of FVs per day, which significantly increased to 2.93 cups at the highest incentive level. Accordingly, a dedicated FV incentive for SNAP shoppers could help to close the gap between current FV consumption and the 4.5 cups of FVs per day recommended by the Dietary Guidelines for Americans. Prior research shows that improving dietary quality results in numerous health benefits for the individual, including a reduced risk of stroke and other cardiovascular diseases, a reduced risk of developing cancer, and a reduced risk of type 2 diabetes [18].

Additionally, adopting FV incentives at FMs could have important implications for addressing nutrition equity among individuals who use SNAP. Previous research suggests that SNAP FM shoppers purchase and consume more FVs than the average SNAP user [19,20], in some cases while spending less (i.e., Hispanic SNAP FM shoppers; [21]). Further study and testing of the highest-level incentive from the current study represents a promising opportunity to increase the purchasing power of individuals who face nutrition inequity (e.g., food insecurity), demonstrating a commitment to healthier food purchasing and consumption at FMs. Om addition, incentive programs have the potential to bring new customers to FMs and bolster FM use among participants [22].

Our data utilize both maximum likelihood and CACE models, an approach arguably underutilized in public health outcome measurement. For example, we were able to identify study outcomes between participants who used and did not use their assigned vouchers. Separating these two groups identified a considerable difference in effect (increased FV consumption by 0.31 cups vs. 0.16 cups). The CACE model provides additional information and complements more traditional methods stemming from medicine (ITT). As interventions are piloted in public health, the administration of the intervention (process) can require substantial efforts separate from the impact of the intervention when used. These data, therefore, demonstrate the potential of the intervention, which may be useful for policymakers and funders and may help program administrators understand differences in impact based on voucher usage. In this study, the voucher redemption rate for the 2:1 level was 53%.

This study has some limitations that are important to note. First, FV consumption was measured using the DSQ, which is a self-reported questionnaire. Future studies should consider using more objective measures of FV consumption, such as skin carotenoid screeners (e.g., Veggie Meter^TM^) to measure participants’ FV consumption, a technology that was cost-prohibitive when this study was conducted. Additionally, this study focused on changes from the baseline incentive and not FV consumption at FMs without an incentive.

## 5. Conclusions

In conclusion, investing in FM incentive programs, which is supported by this study’s findings and widely across the literature, should be prioritized as a public health intervention to increase FV consumption among the food insecure. Specifically, this RCT supports the effectiveness of incentive programs in increasing spending on FVs at FMs and improving the nutrition behaviors of SNAP shoppers. Such programs address the need to increase purchasing power for low-income consumers, such as SNAP participants, enabling the purchase of healthy foods. This is particularly timely as, in recent years, the price of healthy items such as FVs has increased relative to that of unhealthy items [23]. Accordingly, incentive programs such as the one analyzed in this RCT improve the affordability of FVs for program participants. Incentive programs that increase SNAP shoppers’ ability to purchase additional FVs should be part of future policies to support this population, which will create more equitable access for those whose food budgets are otherwise limited.

## Figures and Tables

**Table 1 ijerph-19-07443-t001:** Participant characteristics.

	Frequency	%
Gender	*n* = 2968	
Male	522	18
Female	2446	82
Race	*n* = 2708	
White	1959	72
Black or African American	315	12
Asian/Other Pacific Islander	106	4
American Indian/Alaskan Native	106	4
Other Race	222	8
Ethnicity	*n* = 2825	
Hispanic	515	18
Non-Hispanic	2310	82
Age	*n* = 2895	
18 to 27 years	429	15
28 to 37 years	821	28
38 to 47 years	601	21
48 to 57 years	452	16
58 to 67 years	420	15
68 to 77 years	131	5
78 and above	32	1
Food Insecurity (*n* = 2956)		
Food Insecure	2424	82
Food Secure	532	18
Health Status (*n* = 2956)		
Excellent	266	9
Very Good	828	28
Good	1094	37
Fair	561	19
Poor	207	7
Health Conditions (*n* = 2956)		
Heart Disease	148	5
Diabetes	384	13
High Blood Pressure	680	23

*Note*. Participants could choose not to answer a question.

**Table 2 ijerph-19-07443-t002:** Mean baseline FV intake and expenditures.

	Average
Daily Cups, FV Intake by Gender	
Male	3.27
Female	2.95
Daily Cups, FV Intake by Age	
18 to 27 years	2.93
28 to 37 years	3.03
38 to 47 years	3.12
48 to 57 years	2.98
58 to 67 years	2.91
68 to 77 years	3.03
78 years and above	2.75
Daily Cups, FV Intake Overall	3.00
Monthly FV Grocery Expenditures, All Sources, in USD	153.76

**Table 3 ijerph-19-07443-t003:** SNAP expenditures at farmers’ markets, per transaction and by incentive level.

Incentive Ratio	Dollars
0.4	19.03
0.8	25.30 *^,+^
1	26.87 ^+^
1.5	29.73
2.0	36.28 *^,+^

*Note.* * *p* < 0.05 statistical significance indicated for the difference compared with the preceding incentive level; ^+^
*p* < 0.05 compared with baseline level.

**Table 4 ijerph-19-07443-t004:** Numbers and types of Incentives awarded to study participants from 77 participating farmers’ markets.

	Incentive Level			
Incentive Ratio	Baseline	Level 1 (Moderate)	Level 2 (Highest)	Total
0.4	1199	--	--	1199
0.8	--	1060	--	1060
1	1627	--	1108	2735
1.5	--	1002	--	1002
2	--	--	982	982
Nonmonetary	858	--	--	858
Total monetary	2826	2062	2090	6978

**Table 5 ijerph-19-07443-t005:** Percentages of incentives assigned to study participants from 77 participating farmers’ markets.

Incentive Ratio	% ^a^
0.4	17
0.8	15
1 ^b^	39
1.5	14
2	14
Nonmonetary ^c^	24

^a^ Percentages of incentives awarded add up to 100, after rounding, reflecting that the study design equally yet randomly awarded a financial incentive to each participant. ^b^ The percentage of incentives awarded at the 1.0 level was higher than the other percentage levels because the 1.0 level was both the highest incentive level for the Type A FMs, where the 0.4 incentive was the baseline, and the baseline level for the Type B FMs. ^c^ Nonmonetary incentives were not awarded consistently throughout the study and thus are not included in the calculation of the total percentages awarded.

**Table 6 ijerph-19-07443-t006:** Repeated-measures mixed-effects analysis of outcomes.

FV Consumption (in Cups)	
Intercept	2.77 *
0.4	
0.8	−0.04
1.0	0.00
1.5	0.08
2.0	0.16 *^,a^
Nonmonetary	−0.03
Monthly Grocery Expenditures on FV (log transformation; percent change in FV expenditures)	
Intercept	4.44 *
0.4	
0.8	0.04
1.0	−0.02
1.5	0.05
2.0	0.03
Nonmonetary	−0.01
Household size	0.02 *^,a^

*Note*. * *p* < 0.05 indicates a value different from 0. ^a^ Application of the CACE methodology, for which effects were analyzed only for those participants who spent their incentive, indicated statistically significant increases in FV consumption only at the 2.0 incentive level and only for certain household sizes.

## Data Availability

The data presented in this study are not currently publicly available due to privacy restrictions for human participants.

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
