# Peer review of "Understanding Impacts of SNAP Fruit and Vegetable Incentive Program at Farmers’ Markets: Findings from a 13 State RCT"

_ijerph, 2022, doi:10.3390/ijerph19127443_

Round 1

Author Response

Thank you for taking the time to review our manuscript. We appreciated your thorough comments and believe they substantially improved our manuscript. Please see the attached file for our responses. 

Reviewer 2 Report

If possible consider shortening the title to make it more objective. Make it clear that it is a trial in the title

Leave the introduction in continuous text without subtopics to facilitate the logical reasoning and the understanding of the hypothesis of the study. 

Insert references of other studies in line 58 page 2.

Leave the same objective in the introduction and in the abstract.

Page 3. lines 113-117 How many participants were allocated to each arm of the study?

Page 4. lines 155-164 - was any treatment done to reduce intra- and inter-individual diet variability? 

It makes more sense to put BMI in table 2 and not 3.

Add reference page 11 line 341 and 342 about BMI

Put the limitations of the study at the end of the discussion

It was not clear to me how changes in food prices over time were controlled for in the analyses. Could you please clarify in the methods?

Author Response

Thank you for taking the time to review our manuscript. We appreciate your comments and believe they substantially improved our paper. Please see the attached file for our responses. 
